# Targeted Treatment of Follicular Lymphoma

**DOI:** 10.3390/jpm11020152

**Published:** 2021-02-22

**Authors:** Karthik Nath, Maher K. Gandhi

**Affiliations:** 1Mater Research Institute, University of Queensland, Brisbane, QLD 4101, Australia; karthik.nath@uqconnect.edu.au; 2Department of Haematology, Princess Alexandra Hospital, Brisbane, QLD 4102, Australia

**Keywords:** follicular lymphoma, Bruton tyrosine kinase inhibitors, BCL2 inhibitors, anti-CD47, antibody-drug conjugates, monoclonal antibodies, T-cell engaging bispecific antibodies, chimeric antigen receptor T-cells, immunomodulatory agents, neoantigens

## Abstract

Follicular lymphoma (FL) is the most common indolent B-cell lymphoma. Advanced stage disease is considered incurable and is characterized by a prolonged relapsing/remitting course. A significant minority have less favorable outcomes, particularly those with transformed or early progressive disease. Recent advances in our understanding of the unique genetic and immune biology of FL have led to increasingly potent and precise novel targeted agents, suggesting that a chemotherapy-future may one day be attainable. The current pipeline of new therapeutics is unprecedented. Particularly exciting is that many agents have non-overlapping modes of action, offering potential new combinatorial options and synergies. This review provides up-to-date clinical and mechanistic data on these new therapeutics. Ongoing dedicated attention to basic, translational and clinical research will provide further clarity as to when and how to best use these agents, to improve efficacy without eliciting unnecessary toxicity.

## 1. Introduction

Follicular lymphoma (FL) is the second most common type of non-Hodgkin lymphoma (NHL) in western countries [1]. It is characterized by a waxing and waning disease course where patients can experience multiple relapses between disease-free periods. Although patients with prolonged responses have excellent outcomes, those with early relapse and progression, or histological transformation, can have dismal outcomes [2,3].

Given the marked heterogeneity of the FL disease course, numerous prognostic models to predict treatment outcomes have been evaluated. This includes the follicular lymphoma international prognostic index (FLIPI) [4], which utilizes five clinical risk factors, and the more recently described m7-FLIPI, which incorporates the tumor’s mutational profile in addition to clinical variables [5]. But despite the availability of such risk scores, a biomarker-guided treatment approach remains elusive in FL. Indeed, the FL tumor microenvironment, which plays a critical role in the pathogenesis of this disease, has been shown to be important in predicting outcomes [6,7] and warrants consideration into risk stratification tools.

The introduction of anti-CD20 monoclonal antibodies (mAb) such as rituximab [8], and, more recently the type II anti-CD20 mAb obinutuzumab [9], have revolutionized the management of FL and it is likely that anti-CD20 antibodies will remain at the forefront of the FL treatment armamentarium. But more recent insights demonstrating the intricate relationship between the FL tumor cell and its distinct immune microenvironment have led to a rapid accumulation of novel and innovative targeted therapies that harness the power of the immune system to attack the malignant cell. Moreover, clinical trials utilizing B-cell receptor signaling (BCR) inhibitors, BCL2 inhibitors and specific targeting of FL genetic aberrations, such as *EZH2,* have also demonstrated promising activity (Figure 1 and Table 1). This review summarizes the up-to-date data on the role of novel targeted therapies beyond anti-CD20 mAbs in FL.

## 2. BCR Pathway Inhibitors

Despite the loss of an immunoglobulin (Ig) allele through the hallmark t(14;18) chromosomal translocation with *BCL2* (which is a recurrent feature in >85% of FL) [38], FL tumor cells continue to express surface Ig, highlighting the notion that BCR signaling is necessary for FL cell survival and proliferation. Both antigen-dependent and independent mechanisms have been suggested to promote BCR stimulation [39,40]. This includes a selective pressure of the FL tumor cell to retain surface IgM BCR, which drives stronger BCR signaling than IgG^+^ FL tumor cells [40]. Such IgM^+^ FL tumor cells efficiently bind dendritic cell-specific intercellular adhesion-3-grabbing nonintegrin (DC-SIGN), a C-type lectin receptor present on the surface of both macrophages and dendritic cells. In turn, DC-SIGN induced signaling drives antigen independent, long-lasting BCR aggregation and activation [40]. A subset of FL show BCL2 protein over-expression despite the lack of the t(14;18) [41]. Additionally, *BCL2* mutations, which frequently have enhanced anti-apoptotic activity, are present in a subset of FL (~12%) and are associated with shortened time to transformation and earlier death due to lymphoma. [42].

Upon BCR stimulation, downstream signaling cascades activate membrane-proximal kinases such as spleen tyrosine kinase (SYK), Bruton tyrosine kinase (BTK) and phosphatidylinositol-3-kinase (PI3K). Targeted approaches to disrupt BCR signaling using kinase inhibitors have emerged as an attractive approach in the management of FL (Figure 2).

### 2.1. SYK Inhibitors

Spleen tyrosine kinase (SYK) initiates the BCR signal and is upstream of BTK and PI3K [43]. Irish and colleagues first demonstrated that SYK phosphorylation and BCR-mediated signaling occurred more rapidly in FL tumor B-cells compared to non-tumor B-cells, providing a therapeutic rationale for SYK inhibition in this disease [44]. Fostamatinib disodium, a first-in-class SYK inhibitor, showed only modest activity in a small cohort of recurrent FL with an overall response rate (ORR) of 10% [17]. Entospletinib, a second generation oral SYK inhibitor which is more selective than fostamatinib disodium, has also shown activity in relapsed/refractory (r/r) FL [18].

Notably, co-stimulation of the BCR with IL-4 enhances BCR mediated cellular activation. Indeed, FL tumors are enriched with T-follicular helper cells (TFH) which highly express IL-4 [45], and dual SYK and Janus kinase (JAK) pathway inhibition can simultaneously target BCR and IL-4 respectively [46]. In this regard, initial results from an ongoing phase II study of cerdulatinib, a dual SYK/JAK inhibitor, in combination with rituximab in r/r FL demonstrated an ORR of 59% [19].

### 2.2. BTK Inhibitors

The BTK enzyme is crucial in the B-cell signaling cascade and is essential for B-cell survival and proliferation [47]. Ibrutinib is an orally available, first-in-class irreversible inhibitor of BTK. Bartlett et al. investigated the use of single-agent ibrutinib (560 mg, once daily) in r/r FL [10]. Though reasonably well tolerated, the overall response rate (ORR) was 52.6% amongst patients with rituximab-sensitive disease, but only 16.7% in the rituximab-refractory cohort, and whether this differential response relates to the impact of ibrutinib mediated immunomodulation warrants further consideration [48]. Patients harboring a *CARD11* mutation did not respond to ibrutinib [10] highlighting the predictive utility of the FL genetic profile for targeted treatments. *CARD11* mutations are present in ~12% of FL [5] and gain-of-function mutations result in constitutive activation of NF-κB, (independent of BCR activation) [49].

The phase II DAWN study of single-agent ibrutinib in heavily pretreated FL also demonstrated a modest ORR of 20.9% [11]. Correlative peripheral blood biomarker analyses showed that responders had a significant downregulation in circulating regulatory T-cells (TREG), and increased IFN-γ and IL-12 (T-helper 1 promoting cytokines) compared to non-responders, again suggesting that the ibrutinib response may be associated with beneficial off-target T-cell immunomodulation via inhibition of IL-2-inducible T-cell kinase (ITK) [48]. 

More recently, ibrutinib was investigated in combination with rituximab in frontline FL and demonstrated an ORR of 85% (40% achieving complete response, CR) with a 30-month progression-free survival (PFS) rate of 67% [12]. Ibrutinib-related adverse events included fatigue, diarrhea, nausea, atrial fibrillation, bleeding, and maculopapular rash, but only a minority of patients required dose reductions. The higher ORR in the frontline setting is likely related to the additive effect of rituximab and lower BCR-resistant tumor clones in the absence of prior chemotherapy exposure. Taken together, these encouraging findings suggest that ibrutinib-rituximab could be a therapeutic option in the frontline setting, especially in those not expected to tolerate chemo-immunotherapy. Randomized trials are ongoing to confirm these findings [50]. 

More selective (second-generation) and potent BTK-inhibitors that show promise in other indolent lymphomas are currently being tested. Acalabrutinib was evaluated with rituximab and demonstrated an ORR of 92% in treatment naive FL in a Phase Ib study [13]. Other approaches include combination regimens such as the ongoing phase III SELENE study of ibrutinib with chemotherapy in relapsed FL [51]. A more novel approach is the use of intratumoral CpG (a toll-like receptor 9 agonist) vaccination, local radiation therapy and oral ibrutinib, which recently demonstrated a T-cell mediated systemic anti-tumor response in relapsed FL [52,53].

### 2.3. PI3K Inhibitors

Idelalisib, a PI3Kδ inhibitor, is US Food and Drug Administration (FDA) approved for the treatment of relapsed FL after at least two prior systemic therapies. This first-in-class oral agent was shown to have therapeutic efficacy in a phase II clinical trial of relapsed indolent lymphomas, where 72 patients in this cohort (58% of total) had relapsed FL with refractoriness to both rituximab and an alkylating agent [54]. Patients were treated with 150mg twice daily single-agent idelalisib with therapy maintained until progression. The ORR in FL was 55.6%, (CR, 13.9%), with a median duration of response of 10.8 months [14]. Importantly, a post-hoc analysis demonstrated that idelalisib remained efficacious in FL patients who had progression of disease within 24 months (POD24) of initial chemoimmunotherapy and suggests that idelalisib may be an attractive option in treating high-risk, early relapsing FL [55]. However, the unique toxicity profile of idelalisib is notable. In addition to hematological and gastrointestinal toxicity, clinical vigilance for immune-mediated hepatitis, colitis and transaminitis from idelalisib’s off-target effects on T-cells is required, as is consideration of antimicrobial prophylaxis for opportunistic infections. 

Copanlisib is an FDA approved, intravenous, pan-class PI3K inhibitor that inhibits the catalytic activity of the four PI3K isoforms (PI3Kα, PI3Kβ, PI3Kγ and PI3Kδ), with predominant activity against PI3Kα and PI3Kγ [56]. In a pivotal phase II study of r/r indolent lymphoma (73 of 143 patients with FL) after two or more lines of prior therapy, the ORR to single agent copanlisib was 59%, with a median duration of response of 22.6 months [15]. Interestingly, higher response rates to copanlisib were seen in patients with a high expression of PI3K and BCR pathway genes [15]. Due to the inherent differences in PI3K isoform selectivity compared to idelalisib, copanlisib treatment was associated with transient hyperglycemia and hypertension, but lower immune-mediated toxicity [15]. 

There are numerous other PI3K inhibitors being explored in FL and include duvelisib (FDA approved for r/r FL, PI3Kγ/PI3Kδ inhibitor), umbralisib (PI3Kδ and casein kinase-1ε inhibitor), ME-401 (PI3Kδ inhibitor) and buparlisib (pan-class PI3K inhibitor) [16,57,58,59]. The efficacy of PI3K inhibitors in combination regimens has also been investigated in early phase trials [60].

## 3. Immunomodulatory Agents

Lenalidomide is an orally active immunomodulatory (IMiD) agent that provides pleotropic direct and indirect anti-tumoral effects through modulation of the substrate specificity of the CRL4CRBN E3 ubiquitin ligase in B and T lymphocytes. This results in rapid ubiquitination and degradation of the zinc-finger containing transcription factors Ikaros and Aiolos. Direct tumoricidal effects of lenalidomide are associated with reduced interferon regulatory factor 4, a downstream target of cereblon. Indirect actions are via immune-mediated mechanisms involving stimulation of both the innate (NK-cells) and adaptive immune system (T-cells), modification of microenvironmental cytokines, and inhibition of angiogenesis [61,62,63]. Indeed, lenalidomide was also shown to restore the impaired T-cell synapse with FL tumor cells [64]. Furthermore, IMiD mediated immune modulation enhances the activity of rituximab via NK-cell expansion and subsequent improved antibody-dependent cellular cytotoxicity (ADCC) [65,66].

In this regard, lenalidomide 20mg (for 21-days in 28-day cycles) in combination with rituximab was investigated in the upfront treatment of advanced-stage symptomatic FL in the large, phase III RELEVANCE trial and compared to rituximab plus chemotherapy [20]. Patients in both arms received rituximab maintenance monotherapy for 2 years. There was no significant difference in the rates of CR in patients receiving rituximab-lenalidomide compared to rituximab-chemotherapy (48% vs. 53% respectively), nor was there any difference in 3-year PFS (77% vs. 78%, respectively). As expected, hematological toxicity was more common in those who received rituximab-chemotherapy, but patients on rituximab-lenalidomide had other distinct side effects including higher rates of cutaneous reactions, diarrhea and rash.

The rituximab-lenalidomide combination has also been investigated in relapsed FL in the phase III AUGMENT study, where patients with r/r FL were randomized to receive single agent rituximab versus rituximab-lenalidomide [21]. Notably, rituximab refractory patients were not included in this study. The PFS was significantly higher in rituximab-lenalidomide treated patients compared to rituximab treated patients (39.4 months versus 14.1 months, respectively), albeit with higher rates of treatment emergent adverse events. Interestingly. a post-hoc analysis of the AUGMENT study demonstrated superior efficacy of rituximab-lenalidomide over rituximab in FL patients with POD24 [67]. Based on these studies, lenalidomide has emerged as a promising agent in FL and is now US FDA approved for previously treated FL in combination with rituximab. However, in the absence of a predictive biomarker, selecting the appropriate patient most likely to benefit from an IMiD based therapy remains a challenge. 

Other chemotherapy-free combination approaches using lenalidomide are currently under investigation and yielding encouraging results [68,69,70,71]. Indeed, a CR rate of 92%, with an estimated median 2-year PFS of 96% was reported with obinutuzumab-lenalidomide in the upfront setting [22]. However, triplet combinations of lenalidomide-rituximab and either ibrutinib or idelalisib was associated with significant toxicity [72,73]. Avadomide, a novel immunomodulatory agent, is also being investigated in combination with anti-CD20 antibodies in r/r FL and has shown encouraging results [23,74]. 

## 4. EZH2 and HDAC Inhibitors

EZH2, a histone methyltransferase, supports germinal center (GC) formation and contributes to FL lymphomagenesis [75]. Recurrent gain-of-function mutations in *EZH2* have been reported in ~25% of FL [75], which act to amplify the transcriptional silencing of GC proliferation checkpoint genes and enhance GC B-cell proliferation and accumulation via the formation of an aberrant immunological niche [76,77]. Morschhauser and colleagues investigated the use of tazemetostat, a first-in-class oral inhibitor of mutant and wildtype EZH2 in heavily pre-treated FL [24]. Results from this single arm, phase II clinical trial demonstrated an ORR of 69% in the *EZH2*^mutant^ cohort, with a low prevalence of treatment related adverse events. Tazemetostat is now FDA approved in *EZH2*^mutant^ FL patients in the relapsed setting (after at least two prior systemic therapies). Interestingly, single-agent tazemetostat was also efficacious in the *EZH2*^wildtype^ cohort, albeit with a lower ORR of 35%, but nonetheless highlighting the consistent role of the EZH2 enzyme in GC maintenance and B-cell proliferation [24]. 

More recently, a post-hoc analysis found that early progression of disease was not predictive of response to tazemetostat, suggesting that EZH2 inhibition may be another attractive option for early-relapsing (*EZH2*^mutant^) FL patients [78]. Given as the *EZH2* mutation remodels the FL tumor cells’ interactions with the tumor microenvironment [77], it is intriguing to speculate that combinational approaches of tazemetostat with other targeted therapies may be synergistic. Indeed, the role of tazemetostat in combination with rituximab and lenalidomide is currently being tested [79,80]. Notably, *EZH2* copy-number-gains (CNG) have also been described in ~15% of FL, with such CNG having a similar transcriptional profile to *EZH2* mutations [81], and the role of EZH2 inhibition in those with CNG warrants consideration. 

As PI3K inhibitors (idelalisib, copanlisib, duvelisib) and tazemetostat are both US FDA approved in the 3rd line setting, and in the absence of a head-to-head comparison, selecting the appropriate agent will require an individualized approach. Though caution must be taken when making cross-trial comparisons, intriguingly, tazemetostat was found to have a similar efficacy and lower risk of adverse events compared to PI3K inhibitors when tested using a match adjusted indirect comparison methodology [82]. 

In addition to *EZH* mutations, other chromatin modifying genes (*CREBBP*, *KMT2D*) are frequently mutated in FL [5] and point to potential therapeutic targeting of these FL lesions. However, unlike *EZH2*, mutations in *KMT2D* and *CREBBP* are loss-of-function, and therefore difficult to target [83]. Histone deacetylase (HDAC) inhibitors have been tested in FL, with the ORR ranging from 47% to 64% but with low CRs [84,85]. 

## 5. Venetoclax

FL’s hallmark t(14:18) chromosomal translocation leads to disruption of the *BCL2* oncogene at chromosome 18q21.33, and subsequent over-expression of the anti-apoptotic BCL2 protein. Therefore, selective inhibition of BCL2 with venetoclax in patients with FL should have significant activity in FL. However, response rates with venetoclax monotherapy (and/or combined with anti-CD20 mAbs) have been modest [86]. It is possible that although BCL2 over-expression is a founding event required for lymphomagenesis, the presence of other characteristic mutations in genes such as *CREBBP*, and *KMT2D*, along with heterogenous expression of BCL2, and clonal evolution leading to new genetic aberrations, results in reduce dependence on BCL2 for the FL B-cells survival. Competing pro-survival signals from the tumor microenvironment may also contribute to venetoclax resistance [87]. Combination strategies with other FL therapeutics have been tested to sensitize FL to venetoclax. The phase II CONTRALTO study assessed the addition of venetoclax to chemoimmunotherapy in patients with r/r FL [25]. Over 100 patients were enrolled onto one of three study arms: (1) venetoclax + rituximab (Arm A), (2) venetoclax + bendamustine/rituximab (Arm B) or (3) bendamustine/rituximab (Arm C). The CR rate was modest at 17% in patients receiving venetoclax + rituximab, and efficacy was similar between Arm B (CR, 75%) and Arm C (CR, 69%). However, there was frequent hematologic toxicity in Arm B which led to higher treatment discontinuations compared to Arm C. The potential synergistic activity of venetoclax with other targeted therapies is being actively explored in the frontline setting [71,88].

## 6. CAR T-Cell Therapy

Chimeric antigen receptor (CAR) T-cell therapy has led to a transformational advancement in the management of B-cell malignancies, and is US FDA approved for r/r large B-cell lymphoma after two or more lines of systemic therapy. CAR T-cell receptors are composed of an antigen binding domain, transmembrane domain, co-stimulatory domain and CD3-zeta intracellular signaling domain. The construct is inserted into and expressed on the surface of autologous T-cell using a retroviral or lentiviral vector. In so doing, CAR T-cell constructs cause an anti-tumor effect by expression of high-affinity tumor antigen specific antibodies on autologous T-cells.

Second-generation autologous anti-CD19 CAR T-cell therapies are being evaluated in r/r FL and interim results of the ZUMA-5 study of axicabtagene ciloleucel (axi-cel) for indolent r/r NHL has demonstrated highly promising results [26]. Eighty r/r FL patients (66% of whom had POD24) with at least two prior lines of therapy had an ORR of 95% (CR rate, 80%) to axi-cel with durable responses at early follow-up. The unique toxicity profile of CAR T-cell therapy was similar to that seen with aggressive B-cell lymphomas, with the presence of cytopenias, cytokine-release syndrome (CRS) and immune effector cell-associated neurotoxicity syndrome (ICANS). However, both high-grade CRS and ICANS events were lower than that seen with aggressive B-cell lymphomas [26,89]. 

The planned interim analysis of the ELARA study of tisagenlecleucel (tisa-cel) in r/r FL was also recently reported. Of the 52 patients evaluable for efficacy (60% with a POD24 event), the ORR was 82.7% (CR rate, 65.4%) in the intent-to-treat population, with no high-grade (≥3) CRS and minimal (2%) high-grade neurologic events [27]. 

The high efficacy from these preliminary data inspires cautious optimism regarding the role of CAR T-cell therapy in r/r FL but longer follow-up is necessary to determine durability. The apparent higher efficacy and toxicity seen with axi-cel may be attributed to the CD28 costimulatory domain, which typically causes a rapid T-cell expansion compared to the 4-1BB costimulatory domain of tisa-cel [90]. 

Despite the promise of CAR T-cell therapy, considerable numbers of patients’ relapse and this can be related to antigen loss, limited persistence of the infused CAR T-cell product and immune escape [91,92]. There is a need to better understand and overcome mechanisms of resistance using CAR T-cell therapy and strategies to enhance antitumor activity are being explored in the preclinical setting. This includes *ex-vivo* polarization to IL-9 secreting CAR T-cells, which exert a greater antitumor efficacy compared to standard CAR T-cell constructs [93]. Other approaches to enhance the antitumor efficacy of CAR T-cells include constitutive expression of the CD40 ligand on CAR T-cells [94], modifying CAR T-cells to be used as ‘micro-pharmacies’ for local delivery of tumor suppressor proteins [95], the testing of bivalent CAR constructs that target two domains to mitigate antigen resistance [96], using novel antigen targets [97] and the incorporation of immune checkpoint inhibitors with CAR T-cell treatment [98]. 

## 7. CAR NK-Cells

Natural killer (NK) cells are emerging as a promising cell source for CAR-therapy and are currently being investigated in numerous clinical trials. NK-cells can target tumor cells with decreased or absent expression of major histocompatibility complex (MHC) 1 [99]. Importantly, because NK-cells are human leukocyte antigen (HLA) agnostic, graft-versus-host-disease is not induced. This opens up the option of manufacturing using “off the shelf”, allogenic, umbilical cord blood derived NK-cells (Table 2) [100]. The first-in-human phase 1/2 clinical trial of allogenic CAR NK-cell therapy in B-cell malignancies encoded a CAR against the CD19 B-cell antigen and IL-15 to enhance transduced NK-cell persistence and function [101]. CAR NK-cell administration appeared efficacious and, in keeping with their distinct cytokine profile, was not associated with the development of CRS or ICANS [101]. 

An alternative platform for CAR engineering that also provides off-the-shelf capabilities involves the generation of NK-cells from pluripotent stem cells derived from easily accessible sources such as fibroblasts or peripheral blood [102].

## 8. T-Cell Engaging Bispecific Antibodies

A recent therapeutic approach in B-cell malignancies has been redirecting T-cells to attack the malignant B-cell using bispecific antibodies that bind to a B-cell surface target (e.g., CD20) and a surface component on the T-cell (CD3). The fusion protein generates proximity between the target malignant B-cell and effector T-cell compartment, leading to a synapse formation and T-cell mediated cytotoxicity independent of tumor cell recognition [107]. In this regard, several off-the-shelf IgG based monovalent T-cell engaging bispecific antibodies (TCB) directed against B-cell surface proteins therapies are in advanced clinical development.

Mosunetuzumab is a fully humanized intravenous TCB and is being currently tested in r/r B-cell malignancies. Early results from an ongoing phase I/Ib study which included 62 r/r FL patients (30 of whom had a POD24 event), demonstrated an ORR of 68% (50% achieved CR) [28]. Durable responses were seen in those achieving CR and despite potential safety concerns from potent immune stimulation, rates of high-grade CRS and ICANS were low [28]. Intriguingly, a small subset of FL patients with prior CAR T-cell therapy demonstrated modest responses to mosunetuzuamb, and expansion of previously administered CAR T-cells has been reported [28,108], suggesting a role for TCBs in treating relapses post CAR T-cells.

Odronextamab (REGN1979), another intravenous CD20 TCB used as a continuous therapy showed a robust signal in r/r FL, with a 92.9% ORR (CR 75%) in a phase I study of r/r B-cell NHL [29]. Most frequent high-grade toxicity was predominantly hematological in nature, with only a minority developing high-grade CRS or neurologic adverse events [29]. Baseline CD20 level did not predict response, but CD20 antigen loss was seen in progression and although this is preliminary data, it suggests that treatment emergent resistance to TCBs is partly driven by antigen loss [109]. A phase II trial of odronextamab in r/r B-cell NHL is currently recruiting [110].

Glofitamab, an intravenous novel CD20 targeting TCB with a unique ‘2:1′ molecular configuration (2 CD20 binding domains and 1 CD3 binding domain), was demonstrated to have high potency due to its bivalent binding to CD20 on malignant B-cells in preclinical studies [111]. Preliminary data from a phase I/Ib clinical trial evaluating the efficacy of glofitamab in r/r B-cell NHL was recently presented. Patients received obinutuzumab pre-therapy to effectively mitigate high-grade CRS. Of the 8 efficacy-evaluable FL patients the ORR was 100%, with 75% achieving a CR [30]. The toxicity profile was manageable. Pending updated study results, glofitamab may offer a novel, efficacious and safe approach for the management of heavily pretreated FL patients. A phase I study combining glofitamab with RO7227166, a bispecific antibody like fusion protein that targets CD19 and activates T and NK-cells (via its 4-1BBL costimulatory domain), is currently ongoing [112]. Epcoritamab (GEN3013) is another novel CD20 TCB showing promising anti-tumor activity in r/r FL, and with an acceptable safety profile [31]. A key advantage of epcoritamab is its convenient subcutaneous route of administration.

Collectively, high-grade CRS and neurological toxicity rates appear less pronounced with TCBs than with CAR T-cell therapy [26,89,92]. Another advantage may be their off-the-shelf availability, but the continuous duration of TCB treatment regimens is notable. How to select amongst these CD20 TCBs in r/r FL is a question that remains to be answered, and whether they can be incorporated into other combination strategies with immunotherapeutic partners (such as 4-1BBL) are eagerly awaited.

## 9. Antibody-Drug Conjugates

Antibody-drug conjugates (ADC) are composed of a targeted monoclonal antibody conjugated via a protease-cleavable linker to a cytotoxic agent, such as microtubule inhibitors. Upon cellular binding, the ADC is internalized, broken down and the subsequent release of the microtubule disruptor leads to apoptosis. Polatuzumab vedotin and pinatuzumab vedotin are ADCs delivering monomethyl auristatin E (MMAE) that are conjugated to the B-cell lineage antigens CD79b and CD22, respectively. Polatuzumab vedotin and pinatuzumab vedotin were combined with rituximab and studied in the phase 2 randomized ROMULUS study of r/r FL [32]. The ORR with rituximab-polatuzumab vedotin was 70%, with a CR of 45% and median duration of response of 9.4 months. Patients in the rituximab-pinatuzumab vedotin cohort had an ORR of 62% (CR 5%) with a median duration of response of 6.5 months. Treatment discontinuations from adverse events, which included fatigue, diarrhea, peripheral neuropathy and neutropenia, were notably significant.

Polatuzumab vedotin has been tested in combination with bendamustine-rituximab, but its addition did not appear to improve upon CR rates [113]. More recently, polatuzumab vedotin has been combined with other targeted therapies such as obinutuzumab-lenalidomide [114] and obinutuzumab-venetoclax in r/r FL [115], and early interim analyses demonstrate encouraging ORRs.

## 10. Immune Checkpoint Inhibitors

The FL tumor cells act to re-educate the surrounding tumor microenvironment and create a tumor permissive immune niche [116]. In this regard, the goal of immune checkpoint blockade in FL is to target immunosuppressive ligands and restore an effective anti-tumor immune response. Intratumoral T-cells in FL can express immune-inhibitory receptors, including programmed death-1 (PD-1), T-cell immunoglobulin mucin-3 (TIGIT) and lymphocyte-activation gene 3 (LAG3), leading to an exhausted phenotype [117,118,119,120]. Single agent PD-1 checkpoint blockade was tested in r/r FL but resulted in minimal responses [33]. The absence of PD-1 ligand (PD-L1) gene amplification, as seen in Hodgkin lymphoma, or PD-1 structural variants, as reported in adult T-cell leukemia and diffuse large B-cell lymphoma, may account for these findings [121,122]. Furthermore, Yang and colleagues demonstrated that PD-1 expressing intratumoral CD8^+^ cytotoxic T-cells are functionally active in FL and it is the co-expression of PD-1 with LAG3 that determines an exhausted phenotype [117], and a combination strategy using anti-PD-1 and anti-LAG antibody therapy is under active investigation [123]. Combination approaches with PD-1 checkpoint blockade and anti-CD20 mAb therapy have yielded more favorable results than single agent immune checkpoint blockade [34,35]. Younes et al. tested the combination of a PD-L1 inhibitor (atezolizumab) with obinutuzumab and bendamustine in frontline FL with promising results on interim analysis [124].

## 11. Macrophage Immunomodulation

A recent new class of agent involves induction of pro-inflammatory macrophages to phagocytose FL tumor cells (given in combination with rituximab). Malignant cell surfaces upregulate CD47 [125], which is an innate immune checkpoint that binds to its ligand SIRP to form the CD47-SIRPα signaling complex. SIRPα is a transmembrane protein expressed on myeloid cells (especially macrophages) and signaling induced by the CD47-SIRPα complex inhibits macrophage phagocytosis. Inhibiting the CD47-SIRPα signaling pathway enhances the phagocytosis of tumor cells by macrophages. CD47 blockade can therefore increase macrophage phagocytosis of FL tumor cells by rituximab and also induce a T-cell response by increasing tumor antigen presentation to cytotoxic T-cells [126]. CD47 blockade with Hu5F9-G4 has shown synergistic activity with rituximab in an early phase clinical trial [36] and a phase II study is currently enrolling patients [127].

Other approaches are being investigated. Extracellular adenosine impedes rituximab-mediated cellular phagocytosis by pro-inflammatory macrophages. Pharmacological inhibition of the adenosine 2A receptor as a strategy to overcome adenosine mediated immune evasion, along with other macrophage checkpoint inhibitors in addition to Hu5F9-G4 are currently in development [128,129,130].

## 12. Cancer Vaccines

Neoantigens are novel tumor-specific peptides that are generated by somatic mutations acquired within neoplastic cells. They elicit an anti-tumor T-cell response upon cell-surface presentation by MHC-I/II molecules [131], and because they are only expressed by malignant cells, spare normal tissues. Not only can the study of neoantigens provide detailed mechanistic insight into immune evasion strategies in FL, but they can also serve as highly personalized potential therapeutic targets for the development of tumor-specific vaccines (Table 2). Tumor vaccines targeting neoantigens can be formulated as RNA, DNA, dendritic cell targeting, glycolipid, protein and synthetic long peptide vaccines. Delivery strategies, choice of adjuvants, administration routes and combination with immune checkpoint blockade and/or strategies to evoke immunogenic cell death (such as radiotherapy or oncolytic vaccines) are all factors under active investigation [104,105,106,132]. Patient selection is likely to be important, with FL containing several mutations known to impair antigen presentation [133,134].

Unique to B-cell lymphomas, somatic neoantigens can be derived from V-D-J recombination of the immunoglobulin heavy chain and light chain variable region genes of the malignant FL cells of the B-cell receptor, or from ongoing somatic hypermutation (‘idiotypic neoepitopes’) [135]. Although best described in mantle cell lymphoma using a proteomic based approach [136], idiotypic neoepitopes have also been detected in FL [137]. These neoantigens are mostly derived from framework region 3 and complementarity determining region 3 and are strongly biased towards presentation by MHC-II molecules. Non-immunoglobulin neoantigens have been investigated in early-stage FL using sequencing combined with algorithms based on predicted HLA class I binding efficiencies, which show MHC-I neoantigens derived from frame-shift mutations in commonly mutated genes such as *KMT2D* [138]. Importantly, these neoantigens were typically associated with intact HLA class I presentation.

## 13. Radioimmunotherapy

Radioimmunotherapy (RIT) is a therapeutic approach that uses monoclonal antibodies as a carrier of radionuclides and, in this way, allows for the selective targeting of cancer cell surface antigens [139]. It is well known that FL is a highly radiosensitive tumor. RIT with the radiolabeled anti-CD20 antibody ^90^Y-ibritumomab tiuxetan is FDA approved in r/r FL and as consolidation after upfront induction chemoimmunotherapy. A single center retrospective analysis recently demonstrated the highly efficacious activity of this agent in FL [140]. Furthermore, a prior randomized trial of ^90^Y-ibritumomab tiuxetan had shown higher clinical response rates when compared to rituximab monotherapy in relapsed FL [37]. RIT is generally well tolerated but myelosuppression has been observed, with higher rates seen in patients with significant bone marrow involvement. Although the current practical challenges of RIT administration have limited its clinical application, ongoing advancements in theranostics may lead to its more prominent role within the treatment armamentarium of FL.

## 14. Conclusions

The rapid expansion of novel targeted therapies makes for exciting times in the management of patients with FL. Although a chemotherapy-free future appears to be within our reach, notable challenges remain. Given the biological heterogeneity of FL, it is imperative to be able to identify predictive biomarkers to accurately risk-stratify patients that will likely benefit from therapeutics that utilize particular modes of action [141]. However, with the notable exception of EZH2 inhibitors, this remains to be achieved. There is also the ongoing debate regarding the optimal therapeutic sequencing approach, to both maximize durable response and quality of life. Finally, the most important question in FL is whether targeted therapy offers the possibility of high rates of cure not just in early but also advanced-stage disease [142]. That we can now plausibly ask such questions suggests that the future is bright for the management of FL.

## Figures and Tables

**Figure 1 jpm-11-00152-f001:**
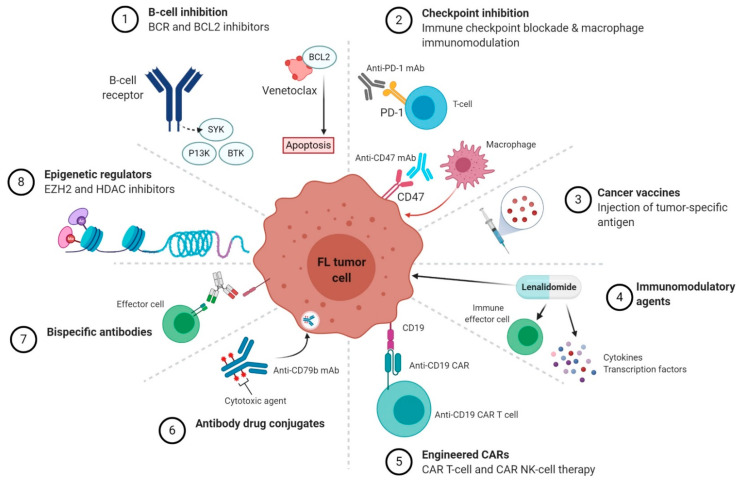
Novel targeted therapeutic classes in follicular lymphoma (FL). Schematic diagram of novel agents grouped according to their mechanism of action. Abbreviations: BCR, B-cell receptor; PI3K, phosphatidylinositol-3-kinase; BTK, Bruton tyrosine kinase; SYK, spleen tyrosine kinase; mAb, monoclonal antibody; PD-1, programmed death-1; CAR, chimeric antigen receptor; NK, natural killer.

**Figure 2 jpm-11-00152-f002:**
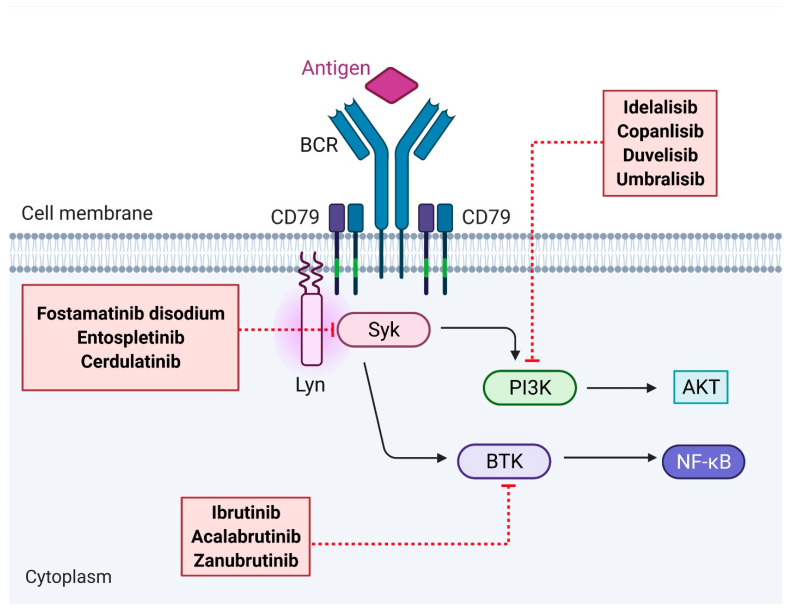
Therapeutic agents targeting the B-cell receptor (BCR) signaling pathway in follicular lymphoma. Abbreviations: BCR, B-cell receptor; Lyn, tyrosine-protein kinase Lyn; SYK, spleen tyrosine kinase; BTK, Bruton tyrosine kinase; PI3K, phosphatidylinositol-3-kinase.

**Table 1 jpm-11-00152-t001:** Summary of targeted treatments in follicular lymphoma. Abbreviations: ORR, overall response rate; CR, complete response; PFS, progression-free survival; -, no additional agent; BTK, Bruton tyrosine kinase; PI3K, phosphatidylinositol-3-kinase; SYK, spleen tyrosine kinase; BR, bendamustine-rituximab; CAR, chimeric antigen receptor; CRS, cytokine-release syndrome; ICANS, immune effector cell-associated neurotoxicity syndrome.

Class	Agent	Combination	Indication	Phase	ORR (CR)	Median PFS	Notable Adverse Events
BTK inhibitors	Ibrutinib	-	Relapsed	II [10]	38% (13%)	14mo	diarrhea, atrial fibrillation bleeding, rash
		-	Relapsed	II [11]	30% (11%)	4.6mo	
		Rituximab	Upfront	II [12]	85% (40%)	42mo	
	Acalabrutinib	Rituximab	Upfront	Ib [13]	92% (31%)		diarrhea
PI3K inhibitors	Idelalisib	-	Relapsed	II [14]	56% (14%)	11mo	hepatitis, colitis, transaminitis, pneumonitis
	Copanlisib	-	Relapsed	II [15]	59% (12%)	11.2mo	Hyperglycemia, hypertension
	Duvelisib	-	Relapsed	II [16]	42% (1%)	9.5mo	diarrhea
SYK inhibitors	Fostamatinib disodium	-	Relapsed	I/II [17]	10%	4.6mo	hematologic toxicity, diarrhea,
	Entospletinib	-	Relapsed	II [18]	17% (0%)	5.7mo	
	Cerdulatinib	Rituximab	Relapsed	II [19]	59% (12%)		lipase increase
Immunomodulatory agents	Lenalidomide	Rituximab	Upfront	III [20]	61% (48%)		cutaneous reactions, diarrhea, hematologic toxicity
		Rituximab	Relapsed	III [21]	78% (34%)	39.4mo	
		Obinutuzumab	Upfront	II [22]	98% (92%)		
	Avadomide	Rituximab	Relapsed	Ib [23]	65% (22%)	6.3mo	
EZH2 inhibitors	Tazemetostat		Relapsed	II [24]	69% (13%)	13.8mo	alopecia
BCL2 inhibitors	Venetoclax	BR	Relapsed	II [25]	84% (75%)		hematologic toxicity
CAR T-cell	Axi-cel	-	Relapsed	II [26]	95% (80%)		CRS, ICANS
	Tisa-cel	-	Relapsed	II [27]	82% (65%)		
T-cell engaging bispecific antibodies	Mosunetuzumab	-	Relapsed	I/Ib [28]	68% (50%)	11.8mo	CRS, ICANS
	Odronextamab	-	Relapsed	I [29]	93% (75%)	12.8mo	
	Glofitamab	Obinutuzumab	Relapsed	I/Ib [30]	100% (75%)		
	Epcoritamab	-	Relapsed	I/II [31]	100% (67%)		
Antibody-drug conjugates	Polatuzumab vedotin	Rituximab	Relapsed	II [32]	70% (45%)	15.3mo	neutropenia
Immune checkpoint inhibitors	Nivolumab	-	Relapsed	II [33]	4% (1%)	2.2mo	immune related adverse events
	Pembrolizumab	Rituximab	Relapsed	II [34]	64% (48%)		
	Pidilizumab	Rituximab	Relapsed	II [35]	66% (52%)	18.8mo	
CD47 blockade		Rituximab	Relapsed	Ib [36]	71% (43%)		anemia
Radio-immunotherapy	^90^Y-ibritumomab tiuxetan	-	Relapsed	III [37]	80% (30%)	11.2mo	myelosuppression

**Table 2 jpm-11-00152-t002:** Ongoing clinical trials of novel immune agents.

Class	Agent	Phase	Population and Study Design
CAR NK-cells	CAR NK-cell	I/II [103]	r/r B-cell lymphoid malignancies Umbilical Cord Blood-Derived CAR NK Cells
Cancer Vaccines	NeoVax	I [104]	Front-line FL Front-line rituximab followed by personalized neoantigen vaccine
	Oncoquest-L-Vaccine	II [105]	Front-line, non-bulky FL Autologous tumor-derived vaccine
	Tumor Vaccine	I [106]	Relapsed FL Personalized tumor vaccine + Nivolumab

Abbreviations: NK, natural killer; r/r, relapsed/refractory; FL, follicular lymphoma.

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
