# Peer review of "Targeted Treatment of Follicular Lymphoma"

_jpm, 2021, doi:10.3390/jpm11020152_

Round 1

Reviewer 1 Report

Nath and Gandhi provide a review of the treatment of follicular lymphoma with a focus on novel targeted therapies and the mechanisms involved. The paper is clearly written and well organized.

It provides a succinct yet appropriate depth of review of the bredth of targeted therapies being pursued.

Comments to consider:

Line 56: consider adding data on the proportion of patients with the t(14;18) translocation (about 85%) as well alternative mechanisms of overexpression of BCL2 as that may relate to future targeted studies as well

Organization: consider that under 2. BCR Path inhibitors  line 55, the specific pathways focused on this (SYK BTK, PI3K) should be subnumered such as with letters rather than as continuous numbering ; also, the order of discussion should mirror your presentation in the 2.0 bcr path inh paragraph to enhance readability

For section 2.0 Consider a figure with the BCR and each of these interconnected pathways to help the reader see where in the chain they fall

Line 94 you note other BTK inh- you should specifically give the data on acalabrutinib as it is approved in some countries already for this indication as well.

Unfortunately, which ritux and obino are noted, there is no mention of radioimmunotherapy. This is approved and effective in foll NHL and many oncologists are unaware of it’s availability. This data should be noted in any discussion of targeted therapies.

Author Response

We thank the reviewer for these encouraging comments.

Line 56: consider adding data on the proportion of patients with the t(14;18) translocation (about 85%) as well alternative mechanisms of overexpression of BCL2 as that may relate to future targeted studies as well

The proportion of patients with t(14:18) has been included (Line 58), and reference (Reference 38). In addition to this, the paragraph on “BCR Pathway inhibitors” has been expanded to highlight mechanisms of antigen independent BCR signalling, BCL2 overexpression in t(14:18) negative cases, and the role of BCL2 mutations in FL.

Organization: consider that under 2. BCR Path inhibitors  line 55, the specific pathways focused on this (SYK BTK, PI3K) should be subnumbered such as with letters rather than as continuous numbering ; also, the order of discussion should mirror your presentation in the 2.0 bcr path inh paragraph to enhance readability

Thank you for highlighting this point. The specific pathways have been re-organized to ensure consistency with the BCR pathway inhibitor pathway paragraph. The new order is as follows: SYK, BTK, PI3K. The specific pathways are now sub-numbered as 2.1 (SYK), 2.2 (BTK) and 2.3 (PI3K).

For section 2.0 Consider a figure with the BCR and each of these interconnected pathways to help the reader see where in the chain they fall

We appreciate the suggestion to include an additional figure with the BCR and each of the discussed interconnected pathways. We have now included a new Figure 2 titled: “Therapeutic agents targeting the BCR signalling pathway”. Figure 2 has been referred to on line 74 of the manuscript text.

Line 94 you note other BTK inh- you should specifically give the data on acalabrutinib as it is approved in some countries already for this indication as well.

Thank you for highlighting this. We have now incorporated a new sentence on lines 150-151 of the main manuscript text to include data on acalabrunitinb. The new reference that this data is presented on is reference number (13). This data has also been referred to in Table 1.

Unfortunately, which ritux and obino are noted, there is no mention of radioimmunotherapy. This is approved and effective in foll NHL and many oncologists are unaware of it’s availability. This data should be noted in any discussion of targeted therapies.

We appreciate the reviewers concern regarding the omission of radioimmunotherapy and agree with the reviewer that many oncologists are unaware of its availability. We have therefore included a new section (section number 13) on radioimmunotherapy in the manuscript text. This involves the addition of 3 new references (references numbers 37, 139, and 140). Radioimmunotherapy is also mentioned in Table 1.

Reviewer 2 Report

This review article from Nath and Gandhi provides a current and appropiate evaluation of therapeutic options both already available in clinic as well as promising new therapies for follicular lymphoma. The manuscript is knowledgeable and easy to read and has the potential to serve as a cornerstone reading for both clinicians and investigators.

The figure is quite ilustrative and serves as an index for the whole manuscript. I miss a table recapitulating the mechanism of action of drugs and clinical trial information including response rates, PFS and toxicity information and also a table highlighting the main characteristics of drugs in preclinical development. 

Also as a minor coment: As a matter of style, I would not use "pola" or "vina", when referring to Polatuzumab vedotin or pinatuzumab vedotin.

Author Response

We thank the reviewer for these encouraging comments.

The figure is quite illustrative and serves as an index for the whole manuscript. I miss a table recapitulating the mechanism of action of drugs and clinical trial information including response rates, PFS and toxicity information and also a table highlighting the main characteristics of drugs in preclinical development.

We thank the reviewer for suggesting the addition of 2 new tables. We have added a new Table 1 which is titled “Summary of targeted treatments in follicular lymphoma” and refer to Table 1 in the main manuscript text on line 50. This includes clinical trial information including response rates, PFS and unique toxicity information.

A new Table 2 titled “Ongoing clinical trials of novel immune agents” and references ongoing clinical trials of CAR NK cell therapy and cancer vaccines. Table 2 is referenced on line 346 and line 467 of the main manuscript text.

Also as a minor comment: As a matter of style, I would not use "pola" or "vina", when referring to Polatuzumab vedotin or pinatuzumab vedotin.

Thanks for pointing this out. The abbreviations “pola” and “pina” have now been removed and replaced with the “polutzumab vedotin” and “pinatuzumab vedotin”.